# Pushing the resolution limit by correcting the Ewald sphere effect in single-particle Cryo-EM reconstructions

Dongjie Zhu [1,2], Xiangxi Wang[1], Qianglin Fang[3], James L Van Etten[4], Michael G Rossmann[3], Zihe Rao[1,5,6] & Xinzheng Zhang[1,5,7]

The Ewald sphere effect is generally neglected when using the Central Projection Theorem for cryo electron microscopy single-particle reconstructions. This can reduce the resolution of a reconstruction. Here we estimate the attainable resolution and report a "block-based" reconstruction method for extending the resolution limit. We find the Ewald sphere effect limits the resolution of large objects, especially large viruses. After processing two real datasets of large viruses, we show that our procedure can extend the resolution for both datasets and can accommodate the flexibility associated with large protein complexes.

[1] National Laboratory of Biomacromolecules, CAS Center for Excellence in Biomacromolecules, Institute of Biophysics, Chinese Academy of Sciences, 100101 Beijing, China. [2] School of Life Science, University of Science and Technology of China, 230026 Hefei, China. [3] Department of Biological Sciences, Purdue University, 240 South Martin Jischke Drive, West Lafayette, IN 47907-2032, USA. [4] Department of Plant Pathology and Nebraska Center for Virology, University of Nebraska, Lincoln, NE 68583-0900, USA. [5] University of Chinese Academy of Sciences, 100049 Beijing, China. [6] Laboratory of Structural Biology, School of Medicine, Tsinghua University, 100084 Beijing, China. [7] Center for Biological Imaging, CAS Center for Excellence in Biomacromolecules, Institute of Biophysics, Chinese Academy of Sciences, 100101 Beijing, China. Correspondence and requests for materials should be addressed to X.Z. (email: xzzhang@ibp.ac.cn)

Recent developments in hardware[1] and improvements in image processing software have increased the resolution of cryo electron microscopy (cryo-EM) single-particle analyses (SPA)[2–7]. Structures of different types of protein complexes ranging in size from 100 kDa to several 1000 kDa have been determined beyond 3 Å resolution[8–18], and in a rare case the resolution has been extended beyond 2 Å[19]. Most of the currently used three-dimensional (3D) reconstruction methods are based on the Central Projection Theorem of weak phase objects[20]. The theorem requires two major assumptions. First, the sample should be thin enough so that the depth of field can be ignored and a single defocus value can be assumed. In cryo-EM, this assumption is equivalent to the assumption that the curvature of the Ewald sphere is neglected[21]. The second assumption is related to the sample being regarded as a weak phase object. Both assumptions are normally valid for small protein complexes that contain mostly light elements. The resolution of a reconstruction has a theoretical limit when the effect of the depth of field or the curvature of the Ewald sphere is ignored, which might have practically limited the attainable resolution of the reconstructions of large protein complexes, e.g., large viruses. Calculation of the resolution limit by different methods has yielded different results[21–23]. These differences probably arise because of different definitions of the resolution. In one method, resolution is determined by comparing the Fourier inversion of a two-dimensional (2D) image with the Fourier inversion of the corresponding 2D projection and then defining the limit of resolution as the frequency where the phase difference reaches 66 degrees[21]. In another method, resolution is defined as the frequency where the phase difference in the contrast transfer function (CTF) caused by the largest variation of the defocus on the object reaches 90 degrees[22]. In a recent publication[23], the resolution limit as a function of the effect of depth of field was estimated by using a threshold of 0.143 in a Fourier shell correlation (FSC) curve calculated between a perfect model and the corresponding map after being convoluted with defocus variation-related 3D point spread function. Since direct comparisons between the resolution limit and the resolution of the reconstruction obtained by cryo-EM SPA were sometimes performed[24, 25], it is better to use the

same resolution definition as is commonly used in SPA where a threshold of 0.143 is used for the FSC curve calculated between the reconstructed maps from two randomly split half datasets. It is equivalent to a threshold of 0.5 used for the FSC curve calculated between a perfect model and a map calculated with all data[26]. Methods have been proposed to break this theoretical resolution limit[21,27–31], but till now, none of them have been successfully applied to SPA of noisy cryo-EM data.

Here we calculate the theoretical resolution limit of the Ewald sphere effect in SPA, and we find currently no SPA reconstructions that exceed this limit. We develop a "block-based" reconstruction method in order to overcome the Ewald sphere effect. This method is tested with two cryo-EM datasets of large viruses and successfully extends the resolution of the reconstructions of two viruses over the theoretical resolution limit.

## Results

**Calculation of the theoretical limit of SPA with Ewald sphere effect.** In conventional reconstruction methods, the thickness of the sample is ignored due to the effect of depth of field, and therefore the resolution limit of the reconstruction is size dependent. In this study, we simulated 3D density models of protein complexes of different sizes and thousands of noise-free cryo-EM images of these protein complexes by taking the depth of field effect into account (Supplementary Information). The reconstructions of these simulated images were calculated using the defocus values measured by the program CTFFIND4[32] and the known orientation information. The resolution of the reconstructed map was then calculated by comparing the map and the corresponding 3D model using a FSC threshold of 0.5. Since the images were noise free and the translation and rotation parameters were error free, this resolution was considered as the theoretical resolution limit caused by the depth of field effect. The resolution limits of different protein complexes were size dependent as shown in Fig. 1a. The simulations showed that the resolution of the reconstruction would be limited to a lower resolution if the images were from a microscope with lower acceleration voltage. Based on these results, a modification from a

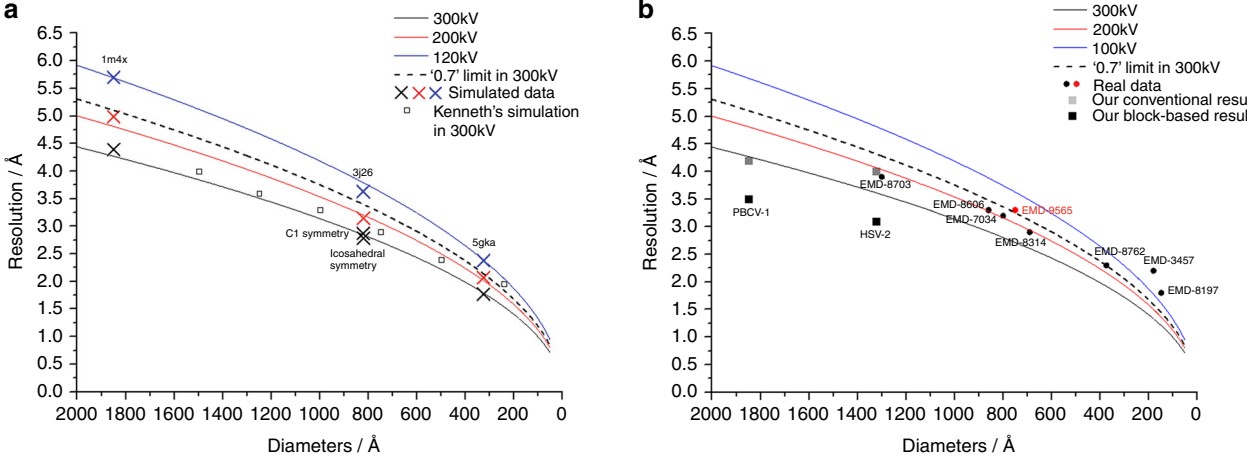

**Fig. 1** The resolution limit caused by the effect of depth of field. **a** Positions marked with a cross (x) represent the calculated resolution limit of protein complexes of variable sizes by using the simulated 300 kV (black), 200 kV (red), and 120 kV (blue) cryo-EM data. The corresponding black, red, and blue lines are the resolution limits calculated using the modified empirical formula. Values for the dashed black line were obtained by applying the DeRoiser's empirical formula that was used as a reference for comparison[21, 22]. The black-edged square is the simulation result performed by Kenneth et al.[23]. **b** Black and red circles with EMDB codes represent the resolutions of typical high-resolution protein complexes generated by cryo-EM SPA at 300 kV and 200 kV, respectively. Black squares represent the resolutions of PBCV-1 virus and HSV-2 virus that were reconstructed by using a block-based reconstruction method. Gray squares represent the resolution of these two viruses that were reconstructed by using a conventional method

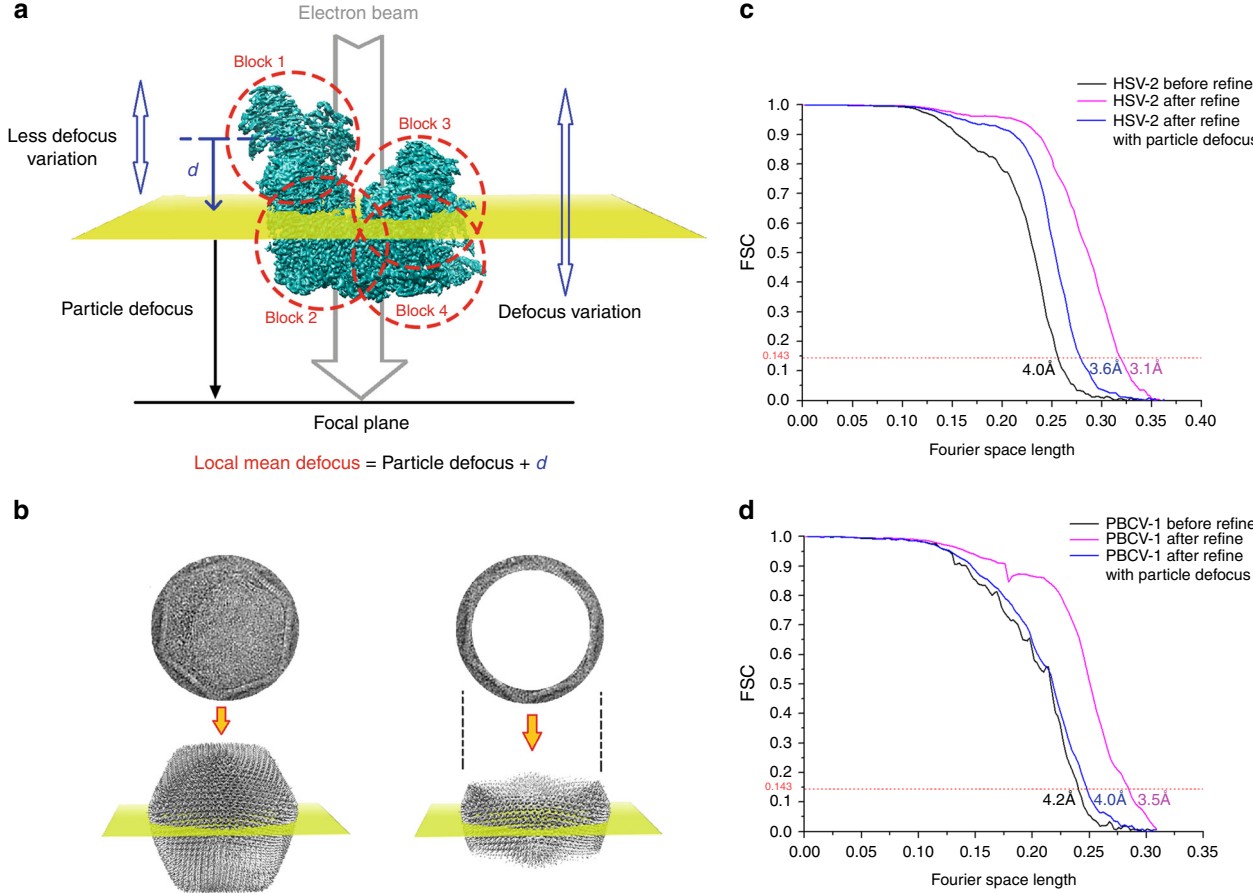

**Fig. 2** The block-based reconstruction, particle defocus, and local mean defoci of blocks. **a** To show how block-based reconstruction works, a density map (EMD-6775) was divided into four blocks, which are circled with red dashes. The distance between the center of mass of the model and the focal plane of objective lens along the Z axis is the particle defocus. Each block has its own local mean focus, which is the sum of the particle defocus values with the distance from the center of the block to the center of the model along the Z axis. **b** The Thon rings calculated from the whole virus (left) or after excluding the central part (right). The region of the virus in the 2D image (up right) corresponds to the part of the virus that is near the particle defocus plane in the 3D virus (down right). **c** The resolutions determined by gold standard FSC at threshold 0.143 of HSV-2 capsid reconstructions by a conventional reconstruction method (colored in black), by block-based reconstruction with local refinement of translational and rotational parameters of each block but without applying local mean defocus (colored in blue), and by block-based reconstruction with local refinement and local mean defocus being applied to overcome the Ewald sphere effect (colored in red). **d** The resolutions determined by gold standard FSC at threshold 0.143 of PBCV-1 virus reconstructions by a conventional reconstruction method (colored in black), by block-based reconstruction with local refinement of translational and rotational parameters of each block but without applying local mean defocus (colored in blue), and by block-based reconstruction with local refinement and local mean defocus being applied to overcome Ewald sphere effect (colored in red)

previous empirical formula[21]:

$$d \approx \sqrt{\frac{2}{t\lambda}} \qquad (1)$$

can be used to estimate the resolution limit for a protein complex having a certain size (Fig. 1), where $d$ is the resolution limit in Fourier space, $t$ is the thickness of the sample, and $\lambda$ is the wave length of the electron. The results showed that the limitation in attainable resolution that we observed after the calculations was higher than the previous calculations (Fig. 1a). Furthermore, non-symmetric objects and viruses with icosahedral symmetry of similar size yield similar resolution limits in our simulation, which indicates that the strong averaging power of the icosahedral symmetry did not help suppress the Ewald sphere effect. Our estimation suggests that the reconstruction (3.3 Å, imaged at 200 kV) of the 66 nm cypovirus capsid shell[25] and the reconstruction (3.9 Å, imaged at 300 kV) of the 120 nm human cytomegalovirus (HCMV) capsid shell[24], which were previously considered to exceed the resolution limit, do not reach the resolution limit that

we estimated (Fig. 1b). In addition, based on the results of our simulation work, and considering the highest resolution of 1.8 Å reported for SPA[19], any protein complexes with their sizes smaller than 30 nm are not normally affected by the Ewald sphere effect.

**The block-based reconstruction method**. To overcome the resolution limit of the conventional SPA reconstruction method, especially for large viruses, we developed a block-based recon-struction method. In this method, any large object with a big defocus gradient can be split into several smaller blocks so that the defocus gradient on each block is much less than that of the whole object (Fig. 2a). Each block can be reconstructed separately using the corresponding 2D signal in the images with its local mean defocus (Fig. 2a). The conventional method for measuring the defocus is to fit the Thon rings of the micrographs with the CTF amplitudes. We define particle defocus as the distance from the center of the mass of this object to the focal plane of the objective lens along the Z axis, the direction of the incident

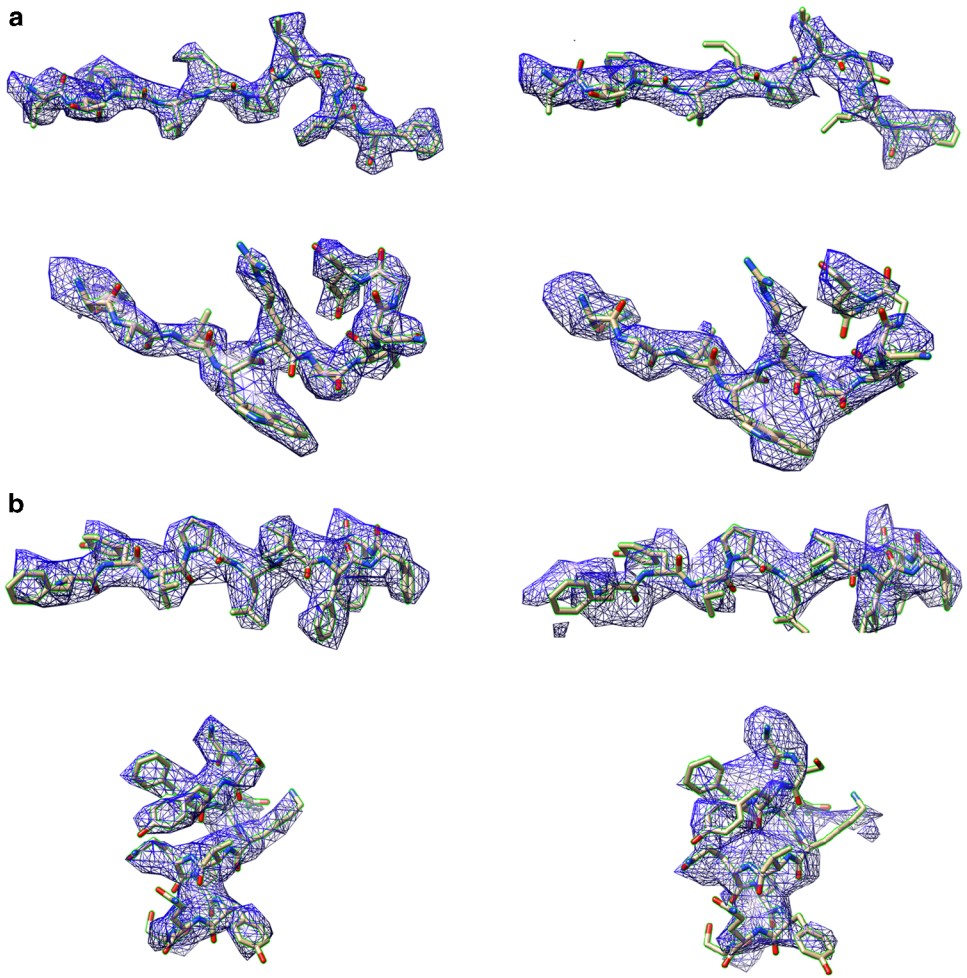

**Fig. 3** Comparison of cryo-EM densities from block-based and conventional reconstruction methods. **a** The densities belong to the major capsid protein in the 3.1 Å resolution HSV-2 reconstruction by block-based reconstruction method (left) and the densities that were from the same areas in the 4.0 Å resolution reconstruction by conventional method (right). The HSV-2 virus structure data are deposited in the Protein Databank with accession code 5ZAP (HSV-2 icosahedral reconstruction after block-based refinement) and in the Electron Microscopy Database with accession code EMD-6907[43]. **b** The densities belong to the major capsid protein Vp54 in the 3.5 Å resolution PBCV-1 reconstruction by the block-based reconstruction method (left) and the densities that were from the same areas in the 4.2 Å resolution reconstruction by the conventional method (right)

electron beam. The difference between measured defocus and particle defocus is <8 nm in most of the cases as shown in Supplementary Figure 2, which indicates that the particle defocus can be represented by the measured defocus. The distance $d$ from the center of each block to the center of mass of the object along the $Z$ axis can be calculated from the rotational parameters of the object provided by the angular parameters of the corresponding image in the previous reconstruction. The local mean defocus of each block is the sum of $d$ and the measured defocus value (Fig. 2a). After reconstructing all the blocks separately, the 3D structure of the complete object was determined by assembling all the reconstructed blocks in real space (see Supplementary Figure 1 and Supplementary Note 1 for detail). This reconstruction method works well for suppressing the depth of field effect in the simulated data.

**Test with two virus datasets**. We then tested this block-based reconstruction method on two cryo-EM datasets. We used the dataset of the herpes simplex virus 2 (HSV-2) virus capsid[33] (1200 Å in averaged diameter) embedded in vitreous ice and that of the *Paramecium bursaria* chlorella virus 1 (PBCV-1)[34] virus (1900 Å in averaged diameter) vitrified on a thin layer of carbon.

For each HSV-2 capsid, we calculated the particle defocus using two different methods—(i) by fitting the Thon rings calculated from the image of the whole virus and (ii) by fitting the Thon rings calculated from the image of the virus that excluded the inner part by a 2D mask (Fig. 2b). In method (ii), only the region of the virus that is near the plane containing the center of mass of the virus and perpendicular to the $Z$ axis was used to calculate the Thon rings (Fig. 2b). Thus the measured defocus value in method (ii) is close to the particle defocus. Our results show that both measurements give similar defocus values (an average error of ~10 nm), which is in agreement with previous simulation studies. We used the defocus from method (ii) as particle defocus. Using conventional reconstruction methods, the entire 1200 Å diameter capsid was reconstructed to a resolution of ~4 Å, which is similar to the resolution of a recently published reconstruction of a similarly sized HCMV capsid[24]. To test the block-based reconstruction method, the 3D asymmetric unit of the icosahedral viral capsid was divided into four blocks. For each block, one cryo-EM image of the virus contains 60 icosahedral symmetry-related projections. Each block could be reconstructed by using all the images each containing 60 symmetry-related projections with their local mean defoci. However, the 4 Å resolution achieved by the conventional reconstruction method was lower than the

predicted theoretical resolution limit, which suggested that the capsid assembly might have certain flexibility that hindered the resolution of the reconstruction. To take the flexibility of the capsid into account, before applying the local mean defocus, the rotational and translational parameters of the 60 projections in each image were refined locally against the corresponding block density. Then the four blocks were reconstructed and combined. The combined reconstruction resulted in a 3.6 Å resolution map that was close to the theoretical resolution limit. The defocus of each projection was then replaced with calculated local mean defocus for refinement. After the four blocks were refined, reconstructed, and combined with icosahedral symmetry being applied, we achieved an overall 3.1 Å resolution map of the whole capsid, which is beyond the attainable resolution limit (Figs. 2c, 1b, and 3a).

In the case of the 1900 Å diameter PBCV-1 virus, the whole virus was reconstructed to 4.6 Å resolution. Since the virus particles were frozen on a thin layer of carbon, a significant portion of the signal of the Thon rings originated from the carbon film that was underneath the sample. Thus the measured defocus did not represent the particle defocus. We were not able to calculate accurate local mean defocus based on the measured defocus. Therefore, after dividing the asymmetric unit of the virus into 11 blocks, the reconstruction with local mean defocus did not improve the map. However, a more accurate particle defocus was obtained by refining the defocus per particle taking into account the depth of field (see Methods). Using the particle defocus, the whole virus was then refined to a resolution of 4.2 Å, which was close to our simulated theoretical resolution limit. For each block, the local mean defocus was recalculated and applied. After the 11 blocks had been refined, combined, and icosahedral symmetry applied, we achieved an overall resolution of 3.5 Å for the whole virus, which is also beyond the theoretical resolution limit (Figs. 2d, 1b, and 3b). Cryo-EM data information of two viruses can be found in Supplementary Table 1. The local resolution of the two reconstructions is shown in Supplementary Figure 3.

## Discussion

We corrected the Ewald sphere effect in large viruses by dividing the object into blocks and reconstructed each block separately with its local mean defocus. We observed that the particle defocus is similar to the measured defocus when the sample is in vitreous ice. A protocol was developed to find the particle defocus when a thin layer of carbon film is present underneath the sample. The accurate determination of particle defocus increases the accuracy of local mean defocus of each block. This ensures an improvement in the resolution. If a proper thickness of ice is achieved in the cryo-EM sample preparation where a single layer of protein complexes is embedded in a thin layer of vitreous ice, then the measured defocus by fitting the Thon ring of the whole micrograph can represent the particle defocus accurately. Therefore, if the resolution of SPA improves further due to the development of better detectors, cryo-EM sample preparation methods, or new phase plate techniques, the Ewald sphere effect on the reconstructions of smaller protein complexes at high resolution might be overcome by using our method. However, enough mass (>500 kD)[17] is required for accurate per particle defocus refinement. Thus our method is not applicable in cases where small protein complexes are randomly distributed in a thick layer of ice with the thickness of ice being much larger than the size of the protein complex. Our block-based reconstruction SPA method can correct the Ewald sphere effect in a manner similar to a 3D-CTF correction, which has been successfully applied in tomographic reconstructions but not in any SPA reconstructions[35~38]. A

refinement against a single block was shown to effectively increase the quality of the map notwithstanding the flexibility among the blocks in the sample. Large protein complexes require large amounts of memory, which can turn out to be a serious computational problem[39]. The procedure described here reconstructs smaller parts of the protein complex separately, which decreases the usage of memory significantly. Low acceleration voltage microscopy can be sufficient for determining high-resolution structures of either large viruses[25] or small protein complexes[40]. Since low acceleration voltage microscopy has a worse depth of field problem than that observed for a 300 kV microscope, the application of the method described here for low acceleration voltage microscopy can turn out to be more advantageous.

We do not subtract the densities above and below the block. The densities above and below the block do contribute noise to the reconstruction. However, the given overlapped densities only overlap with the block in a special orientation. For most of other orientations, the given densities do not overlap with the block and are masked out. Thus the given densities cannot be reconstructed and somewhat contribute to the reconstruction as noise. This kind of noise is not random noise and cannot be averaged out by averaging the images with the same rotational parameters (2D classification) since they have similar overlapped densities. However, the 3D reconstruction combined the 2D images with different rotational parameters and this procedure helped averaging out the overlapped densities. Thus the noise contributed by the overlapped density is weakened by the 3D reconstruction procedure. This is probably why we do not see obvious noise in the reconstruction of the block.

## Methods

**Cryo-EM sample preparation and data collection**. Aliquots of 3.0 μL of PBCV-1 (~5 × 10^11 particles/mL) were applied to a Lacey carbon EM grid. The grids were then blotted for 6 s and flash frozen in liquid ethane using a Gatan CP3 freezer. The frozen grids were loaded on an FEI Titan Krios EM operated at 300 kV equipped with a Gatan K2 Summit detector. Data collection were performed with the Leginon program[41] using a nominal magnification of 18,000 with a dose rate of ~8 e-/(pixel·s) in the "super-resolution" mode resulting in a pixel size of 0.81 Å/pixel. Each micrograph was recorded as a movie composed of 40 frames with an exposure time of 200 ms/frame and a total dose of ~24.4 e-/Å². A total of 5624 such movies were collected.

Aliquots of 3.0 μL of HSV-2 capsid sample with a concentration of ~2 mg/mL were applied to a freshly glow-discharged 400-mesh holey carbon-coated copper grid (C-flat, CF-2/1–2C, Protochips). The grids were blotted for 3.5 s in 80% relative humidity and flash frozen in liquid ethane (Vitrobot; FEI). The frozen grids were loaded on an FEI Titan Krios microscope operated at 300 kV equipped with a Falcon2 camera. Data collections were performed with the Auto-EMation (written by Jianlin Lei) program[41] using a nominal magnification of 59,000 resulting in a pixel size of 1.38 Å/pixel. Each micrograph was recorded as a movie composed of 25 frames with an exposure time of 200 ms/frame and a total dose of ~25 e-/Å².

**Simulation of cryo-EM data considering depth of field**. For a protein complex with a certain size, a 3D model of the protein complex was calculated by EMAN2[3] program e2pdb2mrc.py. The 3D model was first rotated according to the Euler angle for projection and a particle defocus was assigned randomly with the value ranging from 1.0 to 3.0 μm. Then it was divided into several layers along the Z direction with each layer being about 2 nm in thickness. The local mean defocus and the corresponding CTF of each layer were calculated. The projection of each layer was calculated and convoluted with its CTF. The 2D cryo-EM image of the protein complex was simulated by adding all the projections together. After calculating all 2D images, we used CTFFIND4 to measure the defoci of all images.

The simulated cryo-EM images were calculated according to a list of Euler angles, which covered the whole asymmetric unit in the spherical surface with specific steps. A reconstruction was made by the simulated cryo-EM images with the Euler angles and measured defocus. The resolution of the reconstruction was measured by the FSC curve where the correlation between the 3D model and the reconstruction dropped below 0.5. The step size of the Euler angle was selected such that it was fine enough to ensure the corresponding obtained resolution of the reconstruction.

**Particle defocus**. Owing to depth of field, the CTF of a cryo-EM image could not be simply defined by one single defocus value. However, we can still determine the defocus of the protein complex in the cryo-EM image by the conventional method of fitting the Thon ring with CTF. To understand what this fitting result means, the cryo-EM images of protein complexes with different 3D shapes were simulated as mentioned above. The program CTFFIND4 was used to measure the defocus. We found that barring the signal from the vitreous ice (the signal to form Thon ring was mostly from protein)[42], the fitted defocus is close to the defocus of the layer of protein complex that contains its center of mass. For an icosahedral virus with a spherical shape, it is the center of the virus. We call the defocus at the center of mass "particle defocus". The differences between the measured defoci on different simulated images and particle defocus are mostly <8 nm (Supplementary Figure 2).

**Defocus refinement**. For PBCV-1 virus that was vitrified on a thin layer of carbon, a significant part of the signal of the Thon ring came from the carbon, which was underneath the virus. Therefore, the defocus measured by fitting the Thon ring did not represent the particle defocus. Thus the defocus had to be refined during projection match. However, the cryo-EM images were no longer projections of the 3D object because of the Ewald sphere effect. To account for the Ewald sphere effect, one way to perform the projection match was to calculate the projection with Ewald sphere correction as embedded in Frealign or jspr program, which in principle works but has not been proven using real data. Since we used block-based single-particle reconstruction where the protein complex was divided into several blocks, the changes of the defoci among the blocks could be restrained by the distances of the centers of the blocks along the $Z$ axis. Thus the mean defoci of the blocks could be refined in a cooperative manner without Ewald sphere correction as described below:

To refine particle defocus of PBCV-1, we developed a method involving the following steps:

i.  Choose particle defocus search range (normally ~200 nm) and step (normally ~10 nm). Mark the particle defocus value as $PDF_i$.
ii. For particle $i$, using orientation information, calculate defocus variation ($LDFV_{ij}$) between the center of the virus and the center of each block $j$, the local defocus of block $j$:

$$LDF_{ij} = PDF_i + LDFV_{ij} \qquad (2)$$

iii. Calculate the phase residual (PR) between the flip-phased image of the symmetric block and the projection of block,

$$PR_{ij} = PR(LDF_{ij}) = \sum_\omega \frac{\cos(\delta\theta)}{n(\omega)}, \qquad (3)$$

where $1/n(\omega)$ is CTF amplitude weight, $\delta\theta$ is phase difference in Fourier Space within a band of frequency $\omega$.
iv. Summing the $PR_j$ from all blocks in a particle,

$$PR_i = \sum_j PR_{ij} \qquad (4)$$

and for $PDF_i$ in the search range, find the biggest absolute value of the $PR_i$. Choose the corresponding $PDF_i$ as the new particle defocus.

**Combining the blocks**. While combining together the densities of blocks to form the asymmetric unit, we used a simple strategy that kept all densities within a certain radius from the centers of blocks unchanged. A tri-linear interpolation was used to calculate the densities outside the radius based on the distances to the neighboring centers of blocks. To make an icosahedral polyhedron from the asymmetric unit, we calculated the densities of all other symmetry-related points using a tri-linear interpolation function.

**Computational efficiency**. For PBCV-1, we divided the asymmetric unit into 11 blocks. To box the density of each block separately in the micrographs, the centers of the densities that related to blocks in an image of virus were calculated and the corresponding sub-images were extracted with a size of 200×200 pixels, which was large enough to contain a block. The full image size of a whole virus is 1536×1536 pixels. Thus a 3D volume data requires 13.5 GB memory, which becomes a major issue for reconstruction. For instance, the EMAN software cannot handle any map larger than 1290×1290×1290 pixels because of the limitation of the size of array used in EMAN when the programs in EMAN were compiled. Besides, the HSV (1.38 Å/pixel) or PBCV-1 (1.62 Å/pixel) data without binning requires more than 400 and 14 GB of memory per core in RELION[7] 3D refinement and jspr[6], respectively. Such large amounts of memory are not available in most GPU or CPU servers when every core is used to run the job in parallel. After dividing the data into blocks, it required 10 times more storage space than the original particles, but it only requires 0.2% of the memory to process the virus data. Thus the method avoids the major issue related to memory during computation.

**Handedness**. The local mean defocus can be calculated by summing the particle defocus and $d$. If the hand of the reconstruction is wrong, then the local mean defocus can be calculated by summing the particle defocus and subtracting the value of $d$. In the case of wrong handedness, the local mean defocus obtained by summing the particle defocus and $d$ would give a worse resolution map. This phenomenon can be used to determine the handedness of the cryo-EM map in some cases.

**Code availability**. All home-made scripts or programs are available at https://github.com/homurachan/Block-based-recontruction (Supplementary Note 1).

**Data availability**. Densities containing an asymmetric unit of the major capsid protein of PBCV-1 virus have been deposited in the Electron Microscopy Database with accession code EMD-7626. Other data that support the findings of this study are available from the authors upon request.

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

## Acknowledgements

We thank Y. Xiang for discussions and L. Kong for cryo-EM data storage and backup. The project was funded by the National Key R&D Program of China (2017YFA0504700) to X.Z., the Strategic Priority Research Program of CAS (XDB08030204) to X.Z., National Institutes of Health MERIT award (AI011219) to M.G.R., and X.Z. received scholarships from the National Thousand (Young) Talents Program from the Office of Global Experts Recruitment in China.

## Author contributions

D.Z. and X.Z. conceived the idea for this study. X.Z. provided guidance throughout the study. D.Z. and X.Z. generated the simulated datasets and coded the block-based reconstruction program. X.W. and Z.R. provided HSV-2 capsid cryo-EM dataset and Q. F., M.G.R., and J.V.E. provided the PBCV-1 cryo-EM dataset. M.G.R. and X.Z. wrote the manuscript.

## Additional information

**Competing interests:** The authors declare no competing interests.

