## [Peer Review File · Nature Communications]

Editorial Note: This manuscript has been previously reviewed at another journal that is not operating a transparent peer review scheme. This document only contains reviewer comments and rebuttal letters for versions considered at Nature Communications .

REVIEWERS' COMMENTS:

Reviewer #1 (Remarks to the Author):

The revised manuscript addressed most of my previous comments and concerns. However, the revised manuscript still lacks a clear description and the rationale of the 3D reconstruction. It took me quite some time and effort to figure out what exactly authors are doing in calculate the reconstruction of individual blocks, and why it is a valid approach.

I understand why it is not necessary to subtract the density below and above a specific block in this scheme. However, all densities above and below the block would contribute substantial noise to the reconstruction. Giving the fact that the resolution indeed improved, the method obviously works. Giving a clear description would help readers to understand the method better.

Yifan Cheng

Reviewer #2 (Remarks to the Author):

The authors have made a good job answering our questions. I think it was clear that none of the reviewers so a fundamental technical problem in the paper, only that it was "much over sold": I feel satisfied in the way it is presented now, where the applicability limits are quite clearly stated

The description of the method has also greatly improved

REVIEWERS' COMMENTS:

Reviewer #1 (Remarks to the Author):

The revised manuscript addressed most of my previous comments and concerns. However, the revised manuscript still lacks a clear description and the rationale of the 3D reconstruction. It took me quite some time and effort to figure out what exactly authors are doing in calculate the reconstruction of individual blocks, and why it is a valid approach.

The area of each block and the center of the block are defined on the 3D reconstruction map achieved by a conventional SPA reconstruction method. The center of the block on the 2D images can be calculated using the translational and rotational parameters of the 2D images. The densities of the block in the 2D image are centered and clipped to create the sub-image. The particle defocus of each virus can be obtained by fitting the Thon ring signal or by a refinement procedure. The defocus variation between the center of the virus and the center of a block along the Z axis can be calculated. The local defocus of a block is the sum of the defocus variation and the particle defocus. The structure of a block can be reconstructed by sub-images with the rotational parameters and the local defocus. In the sub-image, the densities of a block are normally overlapped with the densities from other blocks. We found the structure of the block can be effectively refined using a local

search of the rotational and translational parameters without density subtraction. However, it is a bad idea to do global refinement, the resolution of the final map would be significantly worse probably due to the overlapped densities. After all the blocks being refined, the densities of these blocks are combined to form a virus. We add this description in the supplementary information and the main manuscript about the whole block-base reconstruction procedure to make it clearer.

I understand why it is not necessary to subtract the density below and above a specific block in this scheme. However, all densities above and below the block would contribute substantial noise to the reconstruction. Giving the fact that the resolution indeed improved, the method obviously works. Giving a clear description would help readers to understand the method better.

The densities above and below the block do contribute noise to the reconstruction. However, the given overlapped densities only overlap with the block in a special orientation. For most of other orientations, the given densities do not overlap with the block and are masked out. Thus, the given densities cannot be reconstructed and somewhat contribute to the reconstruction as noise. This kind of noise is not random noise and cannot be averaged out by averaging the images with the same rotational parameters since they have similar overlapped densities. However, the 3D reconstruction combined the 2D images with different rotational parameters and this procedure helped averaging out the overlapped densities.

Thus, the noise contributed by the overlapped density is weakened by the 3D reconstruction procedure. This is probably why we do not see obvious noise in the reconstruction of the block.

This discussion was added to the manuscript.

Yifan Cheng

Reviewer #2 (Remarks to the Author):

The authors have made a good job answering our questions. I think it was clear that none of the reviewers so a fundamental technical problem in the paper, only that it was "much over sold": I feel satisfied in the way it is presented now, where the applicability limits are quite clearly stated

The description of the method has also greatly improved

We thank the reviewers for their constructive comments about the manuscript and their comments have improved the manuscript.